# Genome-Wide Analysis of the *LRR-RLP* Gene Family in a Wild Banana (*Musa acuminata* ssp. *malaccensis*) Uncovers Multiple Fusarium Wilt Resistance Gene Candidates

**DOI:** 10.3390/genes13040638

**Published:** 2022-04-02

**Authors:** Dulce Álvarez-López, Virginia Aurora Herrera-Valencia, Elsa Góngora-Castillo, Sergio García-Laynes, Carlos Puch-Hau, Luisa Alhucema López-Ochoa, Gabriel Lizama-Uc, Santy Peraza-Echeverria

**Affiliations:** 1Unidad de Biotecnología, Centro de Investigación Científica de Yucatán, Calle 43 No. 130, Colonia Chuburná de Hidalgo, Mérida 97200, Yucatán, Mexico; dulce.alvarez@cicy.mx (D.Á.-L.); vicky@cicy.mx (V.A.H.-V.); sergio.laynes@cicy.mx (S.G.-L.); 2CONACYT-Unidad de Biotecnología, Centro de Investigación Científica de Yucatán, Calle 43 No. 130, Colonia Chuburná de Hidalgo, Mérida 97200, Yucatán, Mexico; elsa.gongora@cicy.mx; 3Centro de Investigación y de Estudios Avanzados del Instituto Politécnico Nacional (CINVESTAV-IPN), Departamento de Recursos del Mar, Unidad Mérida, Km. 6, Antigua Carretera a Progreso, Apdo. Postal 73-Cordemex, Mérida 97310, Yucatán, Mexico; carlos.puch@cinvestav.mx; 4Unidad de Bioquímica y Biología Molecular de Plantas, Centro de Investigación Científica de Yucatán, Calle 43 No. 130, Colonia Chuburná de Hidalgo, Mérida 97200, Yucatán, Mexico; luisa_lopez@cicy.mx; 5Tecnológico Nacional de México/Instituto Tecnológico de Mérida, Av. Tecnológico km. 4.5, Mérida 97118, Yucatán, Mexico; gabriel.lu@merida.tecnm.mx

**Keywords:** banana, disease resistance, genome-wide analysis, LRR-RLP, Fusarium wilt, genetic improvement

## Abstract

Banana is the most popular fruit in the world, with a relevant role in food security for more than 400 million people. However, fungal diseases cause substantial losses every year. A better understanding of the banana immune system should facilitate the development of new disease-resistant cultivars. In this study, we performed a genome-wide analysis of the leucine-rich repeat receptor-like protein (*LRR-RLP*) disease resistance gene family in a wild banana. We identified 78 *LRR-RLP* genes in the banana genome. Remarkably, seven *MaLRR-RLPs* formed a gene cluster in the distal part of chromosome 10, where resistance to Fusarium wilt caused by Foc race 1 has been previously mapped. Hence, we proposed these seven *MaLRR-RLPs* as resistance gene candidates (RGCs) for Fusarium wilt. We also identified seven other banana RGCs based on their close phylogenetic relationships with known LRR-RLP proteins. Moreover, phylogenetic analysis of the banana, rice, and *Arabidopsis* LRR-RLP families revealed five major phylogenetic clades shared by these plant species. Finally, transcriptomic analysis of the *MaLRR-RLP* gene family in plants treated with Foc race 1 or Foc TR4 showed the expression of several members of this family, and some of them were upregulated in response to these Foc races. Our study provides novel insights into the structure, distribution, evolution, and expression of the *LRR-RLP* gene family in bananas as well as valuable RGCs that will facilitate the identification of disease resistance genes for the genetic improvement of this crop.

## 1. Introduction

Banana is a staple food or source of income for more than 400 million people in many developing countries in the tropics. It is the most exported fruit in the world, and more than 145 million tonnes are produced annually, with an estimated economic value of US$52 billion [1]. This production is constantly threatened by several biotic stresses, the most notorious of which, in terms of their global impact, is Fusarium wilt and black Sigatoka diseases, which are caused by the fungi *Fusarium oxysporum* f. sp. *cubense* (Foc) and *Pseudocercospora fijiensis*, respectively [2,3]. Foc infects susceptible banana cultivars throughout the roots, causing lethal vascular wilt, and recently, new species of *Fusarium* causing Fusarium wilt on bananas have been described [4]. Foc race 1 was responsible for the devastation of banana plantations destined for export trade in the first half of the 20th century, which relied on the susceptible cultivar Gros Michel [2,5]. The devastation caused by this race is considered one of the most catastrophic events in the history of plant disease epidemics [6]. Banana production was saved by the introduction of resistant Cavendish cultivars. Currently, Foc Tropical Race 4 (TR4) threatens to wipe out the banana export market that relies on Cavendish cultivars, which are highly susceptible to this Foc race [2,5]. The strategy of using fungicides to control these soil-borne pathogens in the field is ineffective. On the other hand, *P. fijiensis* infects susceptible banana cultivars throughout the stomata, causing leaf necrosis and premature ripening of the fruits [7]. An intense regime of pesticide applications is required to control *P. fijiensis*, up to 66 applications per year in some producing regions, which is expensive and environmentally unsound [7]. This situation has serious implications for the global banana value chain and the environment. Therefore, there is an urgent need to genetically improve the current cultivars that are susceptible to these fungal pathogens.

Sources of resistance for Foc and *P. fijiensis* exist in wild bananas, and resistance gene candidates of the nucleotide-binding site and leucine-rich repeat (*NBS-LRR*) class of resistance (*R*) genes have been isolated [8,9,10,11]. Remarkably, one of these genes, *RGA2* (a.k.a. *RGC2*) [9,10], has been shown to provide immunity in Cavendish against Foc TR4 in the field [12]. Moreover, the genome sequences for some of these wild bananas are already available [13,14], and genome-wide analysis of the *NBS-LRR* gene family has been recently reported [15]. The genome-wide analysis of different classes of *R* genes in bananas should facilitate the identification of novel *R* genes for Foc or *P*. *fijiensis*, which could be used to genetically improve the current susceptible banana cultivars, including the forgotten Gros Michel cultivar.

The knowledge gained in the plant immunity field over the past three decades has shown that disease resistance proteins of the receptor-like protein (RLP) and receptor-like kinase (RLK) classes located at the plasma membrane are involved in the perception of pathogens that colonize the extracellular space and activation of defense responses to halt pathogen spread [16]. This plant immune response has been named extracellularly triggered immunity (ExTI), which can result in programmed cell death at the site of a pathogen attack known as the hypersensitive response. This spatial immunity model explains how RLPs or RLKs perceive extracellular immunogenic patterns (ExIPs) through their external LRR domains to mount an effective and robust immune response. The detection of intracellular immunogenic patterns (InIPs) by cytoplasmic receptors, mainly NBS-LRR genes, leads to intracellularly triggered immunity (InTI), which can also result in a hypersensitive response [17,18].

Foc and *P. fijiensis* are hemibiotrophic pathogens that colonize the xylem vessels and apoplastic spaces of leaf tissue, respectively [2,7]. In the case of Foc TR4, we know that an *R* gene of the *NBS-LRR* class is involved in the immune response of banana [12]; however, there is no information about the identity of the R proteins that activate immune responses against the other races of Foc or *P. fijiensis*. In this sense, the recent discoveries made in tomato and its fungal pathogens *Fusarium oxysporum* f.sp. *licopersici* (Fol) and *Cladosporium fulvum*, which are taxonomically closely related to Fol and *P. fijiensis*, respectively [3,19], can give us some clues about the different classes of R proteins that might be involved in the immunity of bananas to the different races of Foc or *P. fijiensis*. The study of the pathosystem tomato–Fol has uncovered remarkable insights into the identity of R proteins that confer resistance to multiple Fol races. The first tomato *R* gene identified to confer resistance to Fol was *I-2*, which belongs to the *NBS-LRR* class and confers resistance to Fol race 2 [20]. Later, the *I-3* gene, which belongs to the *SRLK* class of *R* genes and confers resistance to Fol race 3, was discovered [21]. Subsequently, it was found that the *I* and *I-7* genes belong to the *LRR-RLP* class of *R* genes; the *I* gene confers resistance to Fol race 1, while *I-7* confers broad-spectrum resistance to different Fol races, including races 1, 2, and 3 [22,23]. These findings show that plant cells use both ExTI and InTI to fend off the invasion of *F. oxysporum*. On the other hand, the pathosystem tomato–*Cladosporium fulvum* reveals a dominant role of the LRR-RLP class of R proteins in providing protection against different races of this fungal pathogen [17]. It has previously been proposed that an *LRR-RLP* gene could be involved in resistance to *P. fijiensis* since the tomato Cf-4 protein that confers resistance to *C. fulvum* race 4 belongs to the LRR-RLP class and is capable of detecting the *P. fijiensis* effector protein MfAVR4, an ortholog of the *C. fulvum* CfAVR4 effector, and mounts a hypersensitive-like response in *Nicotiana benthamiana* and banana leaves [3,24]. Although these remarkable findings point to a banana *LRR-RLP* gene involved in immunity to *P. fijiensis*, to date, this hypothesis has not been proven.

Most *LRR-RLP* genes cloned thus far confer resistance to extracellular pathogens and encode proteins with an extracellular LRR domain and transmembrane spanning domain anchored at the cell membrane [25]. Taking into consideration the relevant role of the *LRR-RLP* class of *R* genes to provide protection to extracellular fungal pathogens related to Foc or *P. fijiensis*, a genome-wide analysis of the *LRR-RLP* gene family in the wild banana *Musa acuminata* ssp. *malaccensis* [13], which is resistant to Foc and *P. fijiensis*, would be of great value to the search for potential *R* genes related to these pathogens. Therefore, in the present study, we carried out a genome-wide annotation of *LRR-RLP* family members in *M. acuminata* ssp. *malaccensis* and determined their structure and location on the chromosomes. Moreover, we inferred their phylogenetic relationships with known R proteins of the LRR-RLP class and with the LRR-RLP families of the model plants *Arabidopsis* and rice, and finally, we analyzed the expression of several members of this family. The insights gained in this study provide a valuable platform for the systematic functional analysis of the *LRR-RLP* gene family in bananas that will facilitate the cloning of resistance genes for the genetic improvement of this crop.

## 2. Materials and Methods

### 2.1. Sequence Retrieval, Annotation of LRR-RLPs and Prediction of Subcellular Localization

The banana genome and proteome sequences v2 of *M. acuminata* ssp. *malaccensis* [26] were obtained from the banana genome hub website (https://banana-genome-hub.southgreen.fr/, accessed on 28 May 2020). Sequences with or without a signal peptide at the N-terminus followed by an LRR domain fused to a transmembrane domain and predicted to be located at the cell membrane were considered LRR-RLPs. These criteria were chosen based on the structure and subcellular localization of known LRR-RLP proteins (Appendix A). To retrieve LRR-RLP proteins, first, we built a hidden Markov model (HMM) profile based on the C3-D region [27] obtained from 20 known LRR-RLP proteins (Appendix A) using the hmmbuild program of the HMMER package v3.3 [28] (http://hmmer.org/, accessed on 28 May 2020) with the default parameter settings. Then, the C3-D HMM profile was used to search the banana proteome using the hmmsearch program of the HMMER package v3.3 with the default parameter settings (E-value cutoff of 0.01). A second HMM search was carried out using the C3-D region of banana LRR-RLPs following the previous procedure. The retrieved sequences were analyzed and annotated using a combination of programs. The SMART [29] (http://smart.embl-heidelberg.de/, accessed on 21 July 2021) and InterProScan v5.54-87 [30] (https://www.ebi.ac.uk/interpro/search/sequence/, accessed on 21 July 2021) programs were used to identify the banana LRR-RLPs using the default parameter settings. The isoelectric point and molecular weight were determined with the Compute pI/Mw program [31] (https://web.expasy.org/compute_pi, accessed on 21 July 2021). Finally, the DeepLoc v1 program [32] (https://services.healthtech.dtu.dk/service.php?DeepLoc-1.0, accessed on 21 July 2021) was used to predict the subcellular localization of banana LRR-RLPs. The banana *LRR-RLP* genes were named with the prefix ‘Ma’ indicating *M. acuminata*, followed by numbers designated based on their positions from top to bottom on the 11 chromosomes.

### 2.2. Gene Structure, Identification of Conserved Protein Motifs and 3D Protein Modeling

The Gene Structure Display Server [33] (GSDS, http://gsds.cbi.pku.edu.cn/, accessed on 1 September 2021) was used to identify exons, introns, and untranslated regions (UTRs) in the *LRR-RLP* genes. Then, the MEME v5.4.1 program [34] (https://meme-suite.org/meme/tools/meme, accessed on 23 August 2021) was employed to identify conserved motifs in the LRR-RLP protein sequences, which were displayed with the GSDS program. Finally, 3D protein modeling was carried out using the SWISSMODEL suite [35] (https://swissmodel.expasy.org/, accessed on 19 January 2022), and 3D models were edited with the PyMol program v2 (https://pymol.org/2/, accessed on 19 January 2022).

### 2.3. Chromosome Location of LRR-RLP Genes

Banana *LRR-RLP* genes were mapped onto banana chromosomes based on their physical position in the *M. acuminata* ssp. *malaccensis* v2 genome sequence. Chromosome location was visualized using the MapGene2Chromosome (MG2C) v2 program [36] (http://mg2c.iask.in/mg2c_v2.0/, accessed on 19 March 2022 ). Physical clustering among *LRR-RLPs* was defined following the criteria of Jupe et al. [37]. To form a gene cluster, the distance between two or more neighboring *LRR-RLP* genes was required to be less than 200 kb.

### 2.4. Multiple Sequence Alignment and Phylogenetic Tree Construction

The Clustal Omega program [38] (https://www.ebi.ac.uk/Tools/msa/clustalo, accessed on 2 February 2022) was used to align known LRR-RLP proteins and banana LRR-RLP RGCs with the default parameter settings. Identical and similar amino acids were indicated with black and gray shading, respectively, using the Color Align Conservation program [39] (https://www.bioinformatics.org/sms2/color_align_cons.html, accessed on 2 February 2022). For phylogenetic tree construction, the C3-F region of LRR-RLP proteins was used as proposed by Fritz-Laylin et al. [40]. Multiple sequence alignments were performed with the C3-F region using the MUSCLE program [41] as part of the MEGA v11.0.11 package [42] (https://www.megasoftware.net/, accessed on 20 March 2022). Then, the maximum likelihood method based on the Jones–Taylor–Thornton (JTT) model implemented in the MEGA package was used for the construction of phylogenetic trees. The nodes were tested by bootstrap analysis with 100 replicates, and trees were edited with the FigTree v1.4.4 program (http://tree.bio.ed.ac.uk/software/figtree, accessed on 20 March 2022). The phylogenetic relationships of banana LRR-RLPs with other plant LRR-RLP families were inferred using 57 and 109 LRR-RLPs from *Arabidopsis* and rice, respectively [40,43,44].

### 2.5. Transcriptomic Analysis of Banana LRR-RLP Genes and RT–qPCR

The expression data reported by Li et al. [45] in Appendix A were used to obtain information about the expression levels of banana *LRR-RLP* genes in response to biotic stress. The heatmap was generated using the Rstudio package v1.4.1106 (https://www.rstudio.com/, accessed on 31 January 2022 ). To assess the expression of some banana *LRR-RLP* RGCs in different tissues, three in vitro plants of *M. acuminata* ssp. *burmannica* accession Calcutta 4 were used to extract total RNA from leaf and root tissues. The in vitro plants were grown on MS medium supplemented with 3% sucrose under a 16 h light/8 h dark regime at 25 °C. Total RNA was extracted using the Norgen Plant/Fungi Total RNA purification kit (Norgen, Thorold, ON, Canada) according to the manufacturer's protocol. Total RNA (5 μg) was used for first-strand cDNA synthesis with SuperScript III reverse transcriptase following the manufacturer's protocol (Thermo Fisher Scientific, Waltham, MA, USA). For RT–qPCR, Applied Biosystems™ SYBR™ Green PCR Master Mix (Thermo Fisher Scientific) was used. Three technical replicates were performed for each independent biological replicate. The banana *25S* gene, amplified with primers reported by van den Berg et al. [46], was selected as an internal standard to normalize the expression of the banana RGCs. Cycling conditions were 95 °C for 10 min, followed by 40 cycles of 95 °C for 15 s, 55 °C for 30 s, and 72 °C for 30 s. The relative expression of banana RGCs was calculated according to the 2^−^^ΔΔCt^ method [47]. RT–qPCR was carried out using the StepOnePlus™ Real-Time PCR System (Thermo Fisher Scientific). Primer sequences are listed in Appendix A.

## 3. Results

### 3.1. Identification of LRR-RLP Genes in the Banana Genome

The C3-D HMM profile was built from 20 known LRR-RLPs (Appendix A) and used to search the *M. acuminata* ssp. *malaccensis* proteome v2 revealed 78 *MaLRR-RLPs* (Table 1). A second HMM search was performed using an HMM profile based on banana LRR-RLPs, but no new LRR-RLPs were retrieved. The 78 *MaLRR-RLP* genes were divided into two groups: the first group comprised 59 members (76%) with a signal peptide motif at the N-terminus, LRR domain, and transmembrane domain (SP-LRR-TM), while the second group comprised 19 members (24%) with the LRR and transmembrane domains (LRR-TM) (Table 1). All 78 MaLRR-RLP proteins were predicted to be located at the cell membrane. The protein length of the 78 MaLRR-RLPs ranged from 275 (MaLRR-RLP54) to 1728 (MaLRR-RLP26) amino acids.

### 3.2. Gene Structure of MaLRR-RLPs and Identification of Conserved Motifs

Structural analysis of the *MaLRR-RLP* gene family showed that the majority of its members, 42 genes out of 78 (54%), were predicted to be intronless (Figure 1), while 21 genes (27%) had just one intron, six genes harbored two introns and five genes had three introns. Four genes harbored a large number of introns: *MaLRR-RLP06* and *MaLRR-RLP69* with eight introns, *MaLRR-RLP72* with nine introns, and *MaLRR-RLP70* with ten introns. Remarkably, one large intron of approximately 17 kb in length was present in both *MaLRR-RLP23* and *MaLRR-RLP27*.

Ten conserved motifs were identified by the MEME program by scanning the 78 MaLRR-RLPs (Figure 2). The majority of the detected motifs were associated with LRRs. We did not find motifs associated with a kinase domain. These data support the annotation presented in Table 1 and illustrate the distribution and conservation of these motifs along with the 78 MaLRR-RLP proteins. Members with similar protein structures clustered in the same phylogenetic clade (Figure 2).

### 3.3. Distribution of MaLRR-RLP Genes on Chromosomes

The *MaLRR-RLP* genes were distributed unevenly on 10 chromosomes and were absent on chromosome 11 (Figure 3). Based on the criteria of Jupe et al. [37], a total of 10 *MaLRR-RLP* gene clusters were found: one cluster on chromosomes 6, 9, and 10; two clusters on chromosomes 2 and 3; and three clusters on chromosome 4. Chromosomes 6 and 10 contained the largest clusters, with seven *MaLRR-RLP* genes each. The 10 gene clusters were located in the distal parts of their corresponding chromosomes. A total of 47 *MaLRR-RLP* genes were organized into clusters representing 60% of the whole set of *MaLRR-RLP* genes. The lowest number of *MaLRR-RLP* genes was found on chromosome 8, with just one member located in the middle of this chromosome.

### 3.4. Banana LRR-RLPs with Potential for Foc Race 1 Resistance Clustered on Chromosome 10

Recently, Ahmad et al. [48] reported the genetic mapping of Foc race 1 resistance in the wild banana *M. acuminata* ssp. *malaccensis*. The authors showed that a single dominant resistance locus is responsible for Foc race 1 resistance and mapped near the distal part of chromosome 10. A region of 360 kb associated with Foc race 1 resistance was identified that contained 165 predicted genes, including 19 putative receptor-like kinases (RLKs). The role of RLKs in plant immunity makes them obvious resistance gene candidates (RGCs) to start the search for a Foc race 1 resistance gene within this group of 19 RLKs. Since our study found seven *MaLRR-RLPs* (*MaLRR-RLP69*, *MaLRR-RLP70*, *MaLRR-RLP71*, *MaLRR-RLP72*, *MaLRR-RLP73*, *MaLRR-RLP74*, and *MaLRR-RLP75*) in the distal part of chromosome 10 (Figure 3), we decided to inspect the 165 predicted genes found by Ahmad et al. [48] in the 360 kb region associated with Foc race 1 resistance. We found that the accession numbers of the seven genes identified as *LRR-RLPs* in the distal part of chromosome 10 corresponded to seven of the 19 *RLK* genes selected by Ahmad et al. [48] as putative resistance genes. Due to this finding, we analyzed these 19 putative RLK proteins in more detail using the SMART, InterProScan, and DeepLoc platforms. We did not find any gene encoding an RLK structure (LRR-TM-Kinase); instead, we found the seven MaLRR-RLPs mentioned above, while the remaining 12 genes encoded proteins with LRR, kinase, or malectin domains (Table 2). In our search for LRR-RLPs in the banana genome sequence v2, several sequences were annotated as RLKs, but they were shown to be LRR-RLPs in our annotations. Considering the role of LRR-RLPs in Fusarium wilt resistance [22,23,49], we proposed these seven *MaLRR-RLPs* as Foc race 1 RGCs; therefore, the number of candidate genes for Foc race 1 resistance in the distal part of chromosome 10 can be reduced from 19 [48] to 7 RGCs. Furthermore, of the seven Foc race 1 RGCs, two sequences, *MaLRR-RLP74* and *MaLRR-RLP75*, showed a similar size and superhelical LRR structure [50] to the Fol resistance proteins I and I-7 from tomato (Table 2 and Figure 4 and Appendix A). Therefore, we proposed that these two *MaLRR-RLP* genes could be a good starting point to search for the gene responsible for Foc race 1 resistance, reducing the complexity of the functional assays in a crop such as banana and speeding up the identification of the resistance gene.

### 3.5. Phylogenetic Analysis of the Banana LRR-RLP Family

The ML phylogenetic analysis of 78 MaLRR-RLP members revealed nine major clades and two phylogenetic singletons (Appendix A). Clades I, III, and IX showed the major expansions with 30, 15, and 11 members, respectively. Several members of the MaLRR-RLP family clustered with known disease resistance proteins (Figure 5). We named each clade containing disease resistance LRR-RLP proteins with the name of a representative member of the clade. Clades I, I-7, and HcrVf2 included just one MaLRR-RLP protein, whereas clade RFO2 contained four closely related MaLRR-RLPs. Interestingly, 12 (67%) out of 18 LRR-RLP disease resistance proteins considered for phylogenetic tree construction clustered in clade I-7 along with just one MaLRR-RLP (*MaLRR-RLP58*). The seven Foc race 1 RGCs formed a phylogenetically closely related group. Considering that one or more of these seven RGCs could be involved in Foc race 1 resistance, it was intriguing that they were positioned next to clade I, where there is a gene (*I*) involved in resistance to *F. oxysporum*. Apart from the Foc race 1 RGCs, we also considered as RGCs those *MaLRR-RLPs* that showed a close phylogenetic relationship with known disease resistance *LRR-RLPs*. Therefore, we found a total of 14 *MaLRR-RLPs* that we proposed to be RGCs (Table 3). Only one MaLRR-RLP protein (MaLRR-RLP76) clustered with CVL2 and Fea2 from *Arabidopsis* and rice, respectively; therefore, this MaLRR-RLP could be the ortholog of these two LRR-RLPs involved in plant development (Figure 5). Moreover, the phylogenetic relationship between the MaLRR-RLP proteins and their counterparts in *Arabidopsis* and rice revealed 11 major clades (Figure 6). Clade I comprises the highest number of LRR-RLPs from *Arabidopsis* and rice, with 34 and 40 sequences, respectively. Notably, two subclades emerged from clade I, one from monocot and one from eudicot plants. Clade I also includes seven disease resistance LRR-RLPs from tomato (ELR, I-7, Cf-2, Cf-4, Cf-5, Cf-9, and Hcr9-4E), one from *Brassica napus* (RLM2), and three from *Arabidopsis* (RLP23, RLP30, and RBPG1). Although clade I showed a major expansion of LRR-RLPs in *Arabidopsis* and rice, which was not the case for bananas, only one MaLRR-RLP clustered in this clade. In total, five clades (I, VII, IX, X, and XI) were shared by banana, rice, and *Arabidopsis* LRR-RLP proteins, suggesting an ancient origin for these phylogenetic clades. Clade VI comprised MaLRR-RLPs belonging to banana only, and this clade showed the largest expansion of MaLRR-RLPs, with 31 sequences.

### 3.6. Expression Profiles of Banana LRR-RLP Genes in Plants Infected with Foc Race 1 and Foc TR4

To investigate the expression profiles of the *MaLRR-RLPs* in response to biotic stress and particularly to Foc, we used expression data from a previous study by Li et al. [45], who reported the transcriptomic profiles of a Cavendish cultivar in response to infection by Foc race 1 or Foc TR4. The Cavendish cultivar is immune to Foc race 1 but susceptible to Foc TR4. We found 18 *MaLRR-RLP* genes present in the 11,412 gene transcripts identified in the Cavendish cultivar roots (Figure 7a). The remaining *MaLRR-RLP* genes, including the seven Foc race 1 RGCs, were not present in this collection of transcripts, probably due to their low abundance in root tissue in the context of the sequencing coverage used by Li et al. [45]. Overall, this set of *MaLRR-RLP* genes displayed a different expression pattern between the Foc race 1- and Foc TR4-infected plants (Figure 7a). The expression of four genes (*MaLRR-RLP16*, *MaLRR-RLP29*, *MaLRR-RLP37*, and *MaLRR-RLP43*) was higher in Foc race 1-infected plants than in those challenged with Foc TR4. Conversely, the expression of three genes (*MaLRR-RLP42*, *MaLRR-RLP58*, and *MaLRR-RLP78*) was higher in Foc TR4-infected plants than in the Foc race 1-infected plants. Interestingly, two RGCs, *MaLRR-RLP58* and *MaLRR-RLP78*, which clustered in clades I-7 and I (Figure 5), respectively, were upregulated in response to Foc TR4 (Figure 7a). The expression of these two RGCs plus a representative gene (*MaLRR-RLP74*) of the seven Foc race 1 RGCs was assessed by RT–qPCR in leaf and root tissues of the wild banana *M. acuminata* ssp. *burmannica* accession Calcutta 4, which is resistant to Foc race 1 and Foc TR4. The expression levels of *MaLRR-RLP58* and *MaLRR-RLP78* were higher in root tissue than in leaf tissue; notably, *MaLRR-RLP78* showed a 26-fold increase, whereas the expression of *MaLRR-RLP74* was higher in leaf tissue than in root tissue (Figure 7b). This low expression of *MaLRR-RLP74* in root tissue might explain the absence of transcriptional data for this gene and others in the study by Li et al. [45].

## 4. Discussion

LRR-RLP receptors are anchored in the cell membrane, where they play a fundamental role in the perception of extracellular immunogenic patterns that trigger effective defense responses to halt pathogen spread. Genome-wide analyses of *LRR-RLP* genes in the model plant *Arabidopsis* have provided valuable insights into their structure, genomic organization, expression, and evolution [40,43,51,52]. Moreover, genome-wide analyses of the *LRR-RLP* family have been reported in rice, poplar, cotton, tomato, and legumes [27,40,44,53,54,55], which may lead to the discovery of novel disease resistance genes for the genetic improvement of these crops. In the present study, we report for the first time the genome-wide analysis of the *LRR-RLP* gene family of bananas. We found 78 *MaLRR-RLPs* in the banana genome, which is within the range of *LRR-RLPs* found in other plant genomes; for example, in *Arabidopsis* and poplar, 57 and 82 *LRR-RLPs* have been identified, respectively [40,53]. The distribution of the *MaLRR-RLPs* in the chromosomes showed that most of them formed gene clusters, indicating evolution by tandem gene duplication. This clustering pattern is consistent with the distributions of *LRR-RLP* and *NBS-LRR* disease resistance genes in the chromosomes of other plant species [27,40,52,53,56]. Disease resistance gene families often expand by gene duplication and are under diversifying selection, which allows new pathogen recognition specificities to emerge [57].

More than 60 years have passed since the introduction of the Foc race 1-resistant cultivar Cavendish as a replacement for the highly susceptible Gros Michel cultivar. The resistance shown by Cavendish against Foc race 1 has been quite robust and durable; however, the molecular identity of the resistance gene has remained unknown. The mapping by Ahmad et al. [48] of the Foc race 1 resistance gene to the distal part of chromosome 10 and identification of 19 putative *RLK* genes in this region was a very important step toward the identification of a Foc race 1 resistance gene. In our investigation of the *MaLRR-RLP* family, we found that seven members of this family mapped in the distal part of chromosome 10, and their accession numbers were the same as seven of the nineteen putative *RLKs* reported by Ahmad et al. [48]. According to our manual annotation, these nineteen predicted open reading frames of the banana genome sequence v2 [26] did not show the configuration of the RLK class of resistance proteins; instead, seven of them were predicted to have the LRR-RLP structure of resistance proteins and located in the cell membrane. Therefore, we considered these seven *MaLRR-RLPs* as the most likely genes associated with Foc race 1 resistance. Interestingly, these seven *MaLRR-RLP* RGCs clustered in a clade located next to the tomato *I* gene clade. This gene encodes an LRR-RLP protein capable of recognizing the avirulent protein SIX4 (secreted in xylem 4) from Fol, which is phylogenetically closely related to Foc race 1 and Foc TR4 [19]. This interaction triggers an immune response that halts Fol spread [58,59]. Homologs to SIX genes have been found in the Foc race 1 genome sequence [19], which raises the possibility that a banana LRR-RLP might be involved in the recognition of a SIX homolog from Foc race 1. Further research will be required to prove this hypothesis. Based on these observations, the number of banana RGCs associated with Foc race 1 resistance on chromosome 10 can be reduced from nineteen [48] to seven. We proposed a good starting point to initiate the functional characterization of these seven RGCs with the *MaLRR-RLP74* and *MaLRR-RLP75* genes, as they showed a similar size and 3D structure to I and I-7 deduced proteins that confer resistance to *F. oxysporum* in tomato. In fact, the identification of the *I* and *I-7* resistance genes followed a similar rational approach to reduce the complexity of the number of RGCs. The *I* gene was mapped in a region of 1 Mb with 98 genes, including three *TIR-NBS-LRR* and one *LRR-RLP* gene. Since the three *TIR-NBS-LRR* genes were associated with viral resistance, the *LRR-RLP* gene was chosen instead as the most likely Fol resistance gene for the functional assay, resulting in the identification of the *I* gene [23]. The *I-7* gene was located in a region of 210 kb with 29 genes, of which 18 were root-expressed, and only 1 of them resembled a resistance gene with an LRR-RLP structure, resulting in the identification of the *I-7* gene in the functional assay [22]. Interestingly, these two Fol resistance genes were resolved in different phylogenetic clades, although both clades were in close proximity. Taken together, the findings of Ahmed et al. [48] and our own findings on the *MaLRR-RLP* family should facilitate the cloning of a Foc race 1 resistance gene. A genetic complementation approach using a banana Foc race 1 susceptible genotype such as Lady Finger will be useful to conduct transformation experiments with the seven *LRR-RLP* RGCs to assess their role in Foc race 1 resistance. Lady Finger has been shown to be a tractable genotype to study the role of antiapoptotic transgenes in Foc race 1 resistance [60]. The cloning of a Foc race 1 resistance gene in banana and the availability of the banana *NBS-LRR RGA2* [12] will open numerous opportunities to “resurrect” the cultivar Gros Michel and bring it back to international trade 60 years after its demise.

In addition to the Foc race 1 RGCs located on chromosome 10, we found seven other *MaLRR-RLPs* that we considered RGCs based on their close phylogenetic relationships with LRR-RLPs with known functions in plant immunity. Interestingly, these known LRR-RLP proteins were not scattered in several clades but formed a few clusters, each containing at least one MaLRR-RLP. The LRR-RLPs of these clades may have a common signal transduction pathway to trigger immunity, causing them to cluster together in the phylogenetic tree. In bananas, an RGC of the NBS-LRR class of disease resistance genes known as *RGA2* (a.k.a. *RGC2*) clustered in a phylogenetic clade shared by the tomato *I-2* and melon *Fom-2* genes that confer resistance to *F. oxysporum* [10]. Functional analysis of *RGA2* showed that this gene was capable of mounting an effective immune response against Foc TR4 in the field [12]. This finding supports the idea that the MaLRR-RLPs that clustered with known LRR-RLPs involved in disease resistance may have a role in immunity. Therefore, these *MaLRR-RLP* RGCs represent a valuable platform on which to perform functional assays that may lead to the identification of novel resistance genes to combat the most damaging diseases of bananas, such as Fusarium wilt and black Sigatoka diseases. New technologies based on cisgenesis or gene editing should facilitate the introduction of new disease resistance traits into banana cultivars using resources from the same banana germplasm.

To analyze the phylogenetic relationships of the banana LRR-RLP family members with the complete set of LRR-RLPs from other plant genomes, we chose the LRR-RLP families of the model plants *Arabidopsis* and rice. We found five clades that were shared by banana, rice, and *Arabidopsis*, suggesting an ancient origin for these phylogenetic clades that may predate the divergence of monocot and eudicot plants that occurred approximately 200 million years ago [61]. Moreover, clade I showed the major expansion and diversification of LRR-RLPs from *Arabidopsis* and rice, unlike banana, which showed a radical contraction of members in this clade with just one LRR-RLP protein. Conversely, clade VIII showed a major expansion and diversification of LRR-RLPs only from bananas. The contraction or expansion of LRR-RLPs in these phylogenetic clades may be the result of different selection pressures from microbes that shape the immune receptor diversity in these plant species after having separated from the common ancestor [62,63].

Transcriptional analysis of the *MaLRR-RLP* family in response to biotic stress showed that the expression of 18 genes was present in banana root tissue, and a few genes were upregulated in response to Foc race 1 or Foc TR4, suggesting the presence of cis-acting regulatory elements in the promoters of these genes that respond to biotic stress signals. A recent transcriptional study of the *LRR-RLP* family in *Arabidopsis* showed that biotic stresses perturbed the expression of the largest proportion (87%) of 51 *AtLRR-RLP* genes [51]. For example, the fungal pathogen *Erysiphe orontii* caused an upregulation of 50% of the *AtLRR-RLP* family members, indicating that a significant number of *AtLRR-RLPs* can respond to a particular pathogen [51]. We detected the expression of 23% of the 78 *MaLRR-RLPs* in the transcriptomic data of Li et al. [45]; this percentage probably represents the *MaLRR-RLP* genes with the most abundant transcripts in root tissue, while the remaining genes were not detected due to their low expression levels that escaped detection at the sequencing coverage used. Our RT–qPCR results point in this direction, since the expression of the RGCs *MaLRR-RLP58* and *MaLRR-RLP78* in root tissue was shown to be higher than that of Foc race 1 RGC *MaLRR-RLP74*, whose expression was not present in the transcriptomic data of Li et al. [45]. The upregulation of the RGCs *MaLRR-RLP58* and *MaLRR-RLP78* in root tissue challenged with Foc TR4 was intriguing since these two genes grouped with the Fusarium wilt resistance genes *I-7* and *I* genes, respectively. Further expression analysis of the *MaLRR-RLP58* and *MaLRR-RLP78* genes in Foc TR4-resistant and Foc TR4-susceptible banana genotypes will provide valuable insights into their role in Foc resistance. The increased expression of *MaLRR-RLP78* in root tissue in comparison to leaf tissue indicates that it may function as a root-specific receptor. Overall, this transcriptomic analysis has provided a foundation for further expression and functional studies of the *LRR-RLP* family in bananas.

## 5. Conclusions

In this study, we identified 78 *MaLRR-RLP* genes that were resolved in nine phylogenetic clades, and 60% of them were organized into gene clusters on six chromosomes. Phylogenetic analysis of banana, rice, and *Arabidopsis* LRR-RLP families revealed five major phylogenetic clades shared by these plant species, suggesting an ancient origin that may predate the monocot-eudicot split. Furthermore, evidence of expansion and contraction of *MaLRR-RLP* gene numbers in particular phylogenetic clades was found. In addition, our data analysis showed that seven *MaLRR-RLPs* formed a gene cluster in the distal part of chromosome 10, where resistance to Foc race 1 has been previously mapped. Therefore, we considered these seven *MaLRR-RLPs* as RGCs for Foc race 1. We also identified seven other *MaLRR-RLP* RGCs based on their close phylogenetic relationships with known *LRR-RLPs*. Finally, we detected the expression of several members of this family in transcriptomic data of plants treated with Foc race 1 and Foc TR4, where some *MaLRR-RLP* genes were shown to be upregulated in response to these Fol races. Taken together, the comprehensive analysis of the *MaLRR-RLP* family in a wild banana provides valuable resources that will facilitate the cloning of a Foc race 1 resistance gene and other disease resistance genes for the genetic improvement of banana cultivars and perhaps other crops.

## Figures and Tables

**Figure 1 genes-13-00638-f001:**
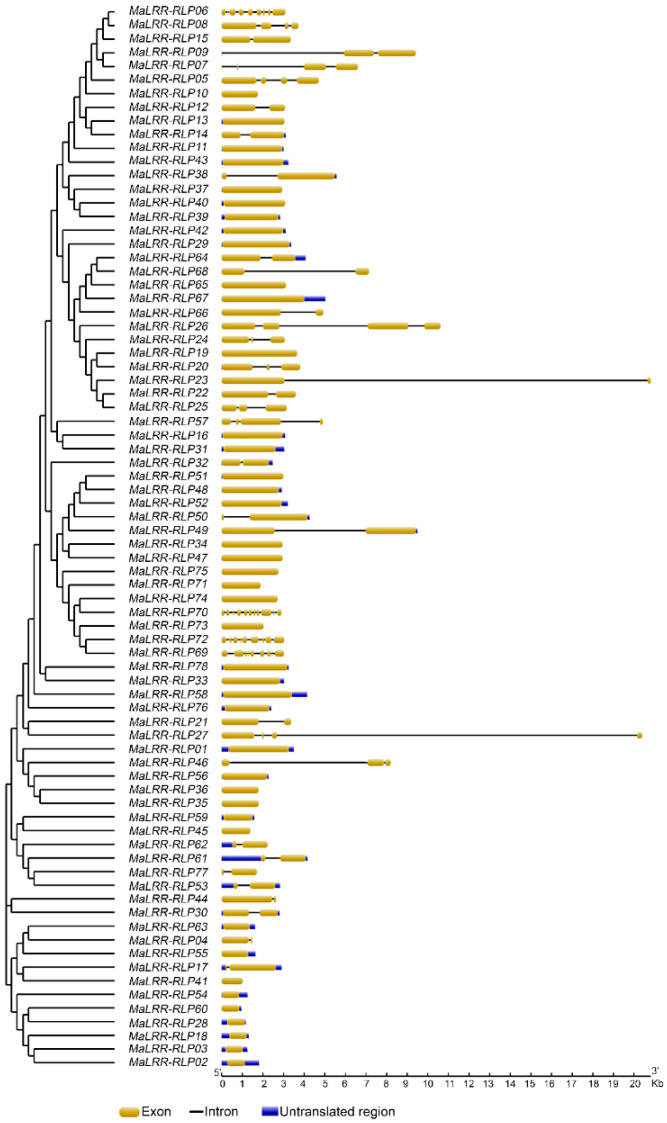
Gene structure of banana *LRR-RLP* genes. Yellow boxes indicate exons, blue boxes indicate untranslated regions and black lines represent introns.

**Figure 2 genes-13-00638-f002:**
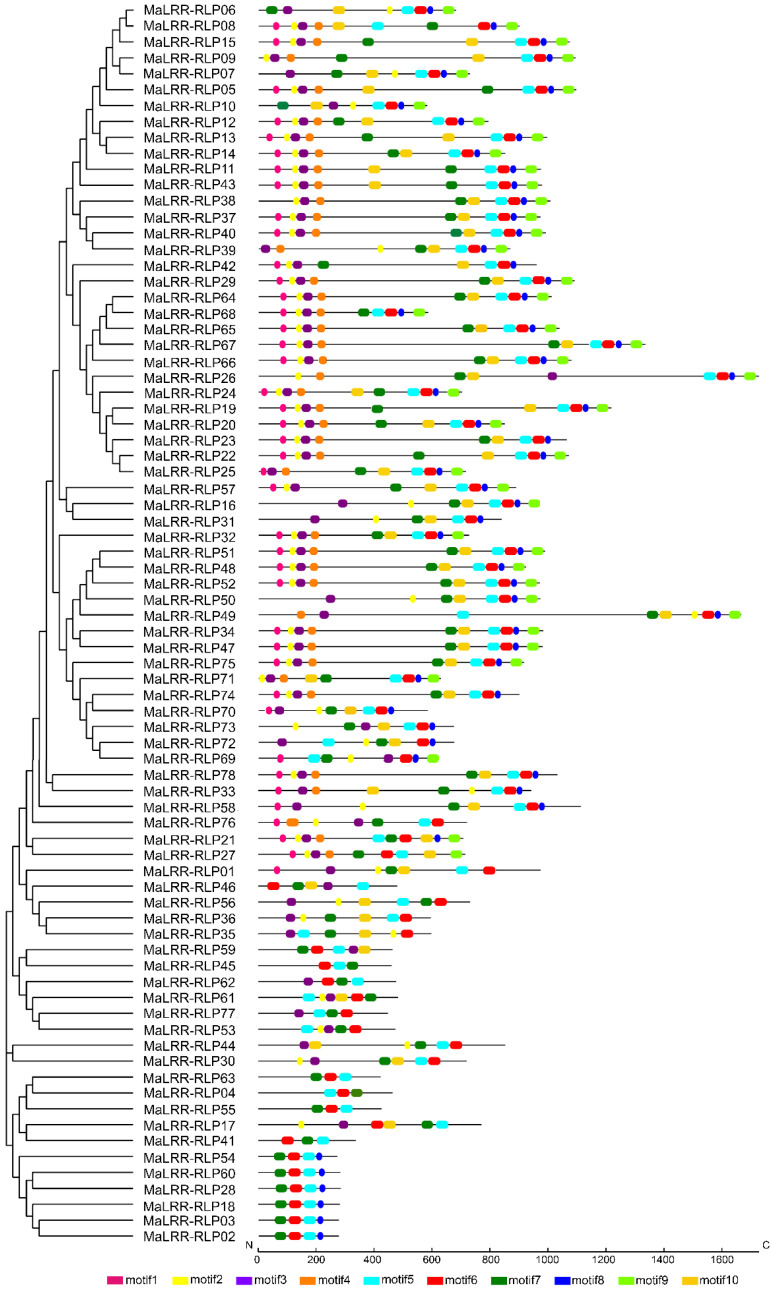
Conserved motifs of banana LRR-RLP proteins. The number of conserved motifs identified by the MEME program was set to 10.

**Figure 3 genes-13-00638-f003:**
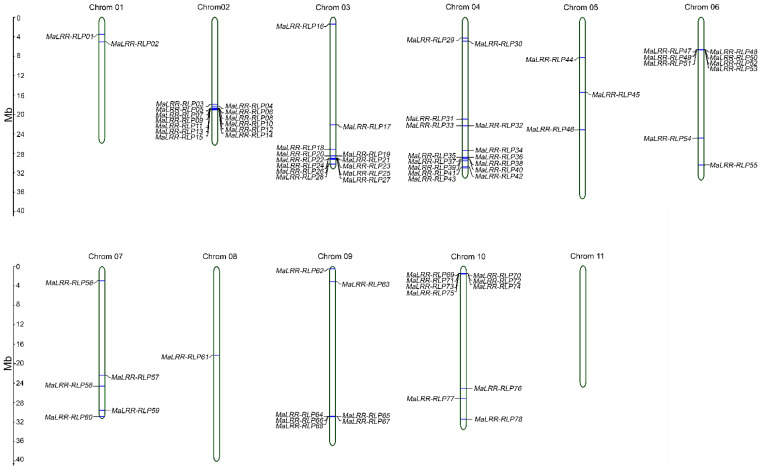
Chromosomal distribution of banana *LRR-RLP* genes. The chromosome number is shown above each chromosome. The left scale represents the length of chromosomes in megabases.

**Figure 4 genes-13-00638-f004:**
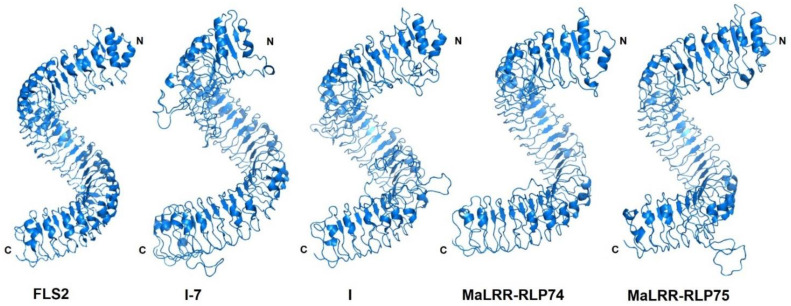
Cartoon view of the predicted LRR structures of I-7, I, MaLRR-RLP74 and MaLRR-RLP75. *I-7* and *I* confer resistance to Fusarium wilt in tomato, and RGCs *MaLRR-RLP74* and *MaLRR-RLP75* clustered in the distal part of chromosome 10, where resistance to Foc race 1 has been mapped [48]. Crystal structure data of *Arabidopsis* FLS2 (PDB Acc. No. 4mn8.1) were used for the model constructions. The structure of FLS2 LRR is superhelical [50].

**Figure 5 genes-13-00638-f005:**
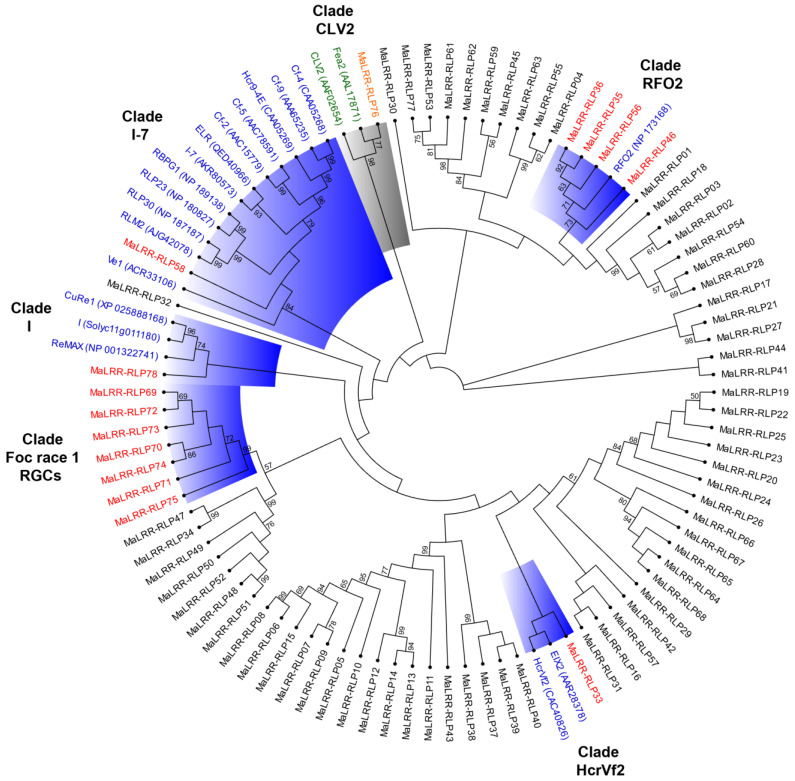
Maximum likelihood phylogenetic tree of the banana LRR-RLP family and LRR-RLPs from different plant species involved in disease resistance or development. The names and accession numbers (in parentheses) of LRR-RLPs involved in plant immunity or development are highlighted in blue or green, respectively. Banana LRR-RLP resistance gene candidates or with a potential role in development are highlighted in red or orange, respectively. Clades containing MaLRR-RLP RGCs are highlighted in blue. Numbers on the branches indicate the percentage of 100 bootstrap replications supporting the particular nodes, and only those ≥50% are shown. The C3-F region of LRR-RLP proteins was used for phylogenetic tree construction.

**Figure 6 genes-13-00638-f006:**
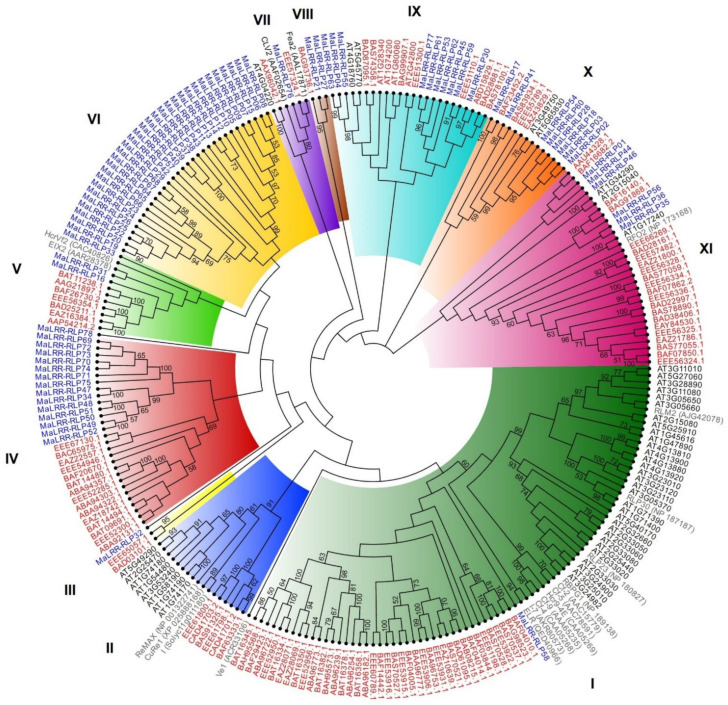
Maximum likelihood phylogenetic tree of banana, rice, and *Arabidopsis* LRR-RLP families. Banana, rice, and *Arabidopsis* LRR-RLP members are highlighted in blue, red, and black, respectively. The names and accession numbers (in parentheses) of LRR-RLPs involved in plant immunity or development are highlighted in gray or green, respectively. Numbers on the branches indicate the percentage of 100 bootstrap replications supporting the particular nodes, and only those ≥50% are shown. The C3-F region of LRR-RLP proteins was used for phylogenetic tree construction.

**Figure 7 genes-13-00638-f007:**
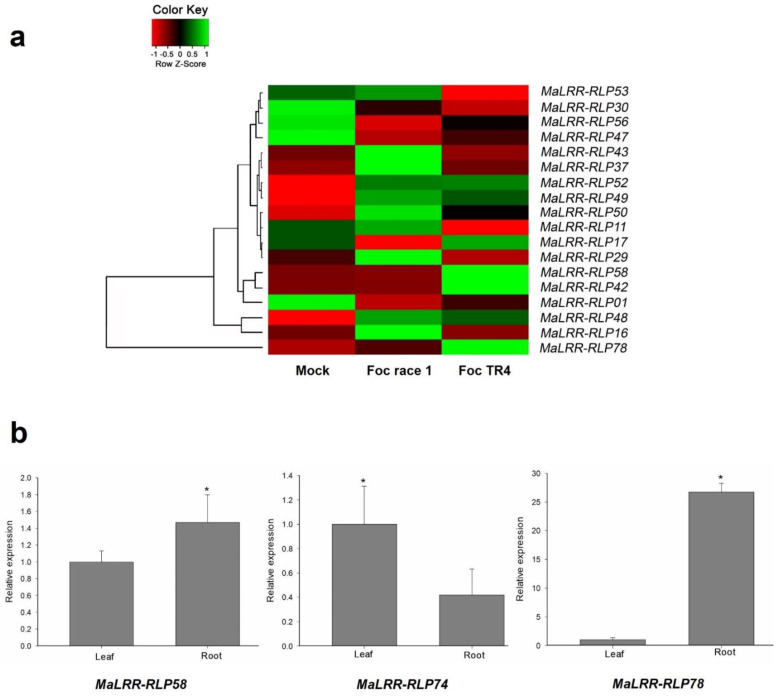
Expression profiles of banana *LRR-RLP* genes. (a) Expression profiles of *MaLRR-RLP* genes in response to Foc race 1 or Foc TR4 at 51 h postinoculation. The color scale reflects gene expression levels. Expression data reported by Li et al. [45] were used to obtain information about the expression levels of *MaLRR-RLPs* in response to Foc race 1 or Foc TR4. (b) RT–qPCR of *MaLRR-RLP* RGCs *MaLRR-RLP58*, *MaLRR-RLP74*, and *MaLRR-RLP78* in leaf and root tissues of a *M. acuminata* ssp. *burmannica* in vitro. The relative expression of genes was calculated by the 2^−ΔΔCT^ method [47]. The mean ± S.D. of three biological replicates is presented. Asterisks indicate *p* ≤ 0.05 (*t* test).

**Table 1 genes-13-00638-t001:** Banana *LRR-RLP* genes identified in this study.

Gene Name	Sequence ID	Domains ^a^	Protein Length	kDa	pI	Subcellular Location
*MaLRR-RLP01*	Ma01_p05590.1	SP-LRR-TM	976	104.65	6.27	Cell membrane
*MaLRR-RLP02*	Ma01_p07780.1	SP-LRR-TM	280	29.80	7.60	Cell membrane
*MaLRR-RLP03*	Ma02_p10510.1	SP-LRR-TM	280	29.77	6.71	Cell membrane
*MaLRR-RLP04*	Ma02_p11370.1	SP-LRR-TM	465	49.35	9.33	Cell membrane
*MaLRR-RLP05*	Ma02_p12110.1	SP-LRR-TM	1098	123.51	5.75	Cell membrane
*MaLRR-RLP06*	Ma02_p12120.1	LRR-TM	684	76.37	5.00	Cell membrane
*MaLRR-RLP07*	Ma02_p12140.1	LRR-TM	732	82.10	6.08	Cell membrane
*MaLRR-RLP08*	Ma02_p12150.1	SP-LRR-TM	903	102.55	5.98	Cell membrane
*MaLRR-RLP09*	Ma02_p12160.1	LRR-TM	1096	123.30	5.30	Cell membrane
*MaLRR-RLP10*	Ma02_p12190.1	LRR-TM	584	65.46	5.73	Cell membrane
*MaLRR-RLP11*	Ma02_p12580.1	SP-LRR-TM	976	108.33	5.75	Cell membrane
*MaLRR-RLP12*	Ma02_p12610.1	SP-LRR-TM	795	89.78	5.55	Cell membrane
*MaLRR-RLP13*	Ma02_p12630.1	LRR-TM	998	113.02	5.96	Cell membrane
*MaLRR-RLP14*	Ma02_p12660.1	SP-LRR-TM	853	96.55	5.71	Cell membrane
*MaLRR-RLP15*	Ma02_p12670.1	SP-LRR-TM	1076	120.27	6.48	Cell membrane
*MaLRR-RLP16*	Ma03_p02400.1	SP-LRR-TM	973	106.68	5.15	Cell membrane
*MaLRR-RLP17*	Ma03_p19590.1	SP-LRR-TM	772	84.27	6.44	Cell membrane
*MaLRR-RLP18*	Ma03_p27010.1	SP-LRR-TM	281	29.90	5.88	Cell membrane
*MaLRR-RLP19*	Ma03_p28990.1	SP-LRR-TM	1219	134.93	5.40	Cell membrane
*MaLRR-RLP20*	Ma03_p29020.1	SP-LRR-TM	851	95.65	5.71	Cell membrane
*MaLRR-RLP21*	Ma03_p29780.1	SP-LRR-TM	709	78.56	5.25	Cell membrane
*MaLRR-RLP22*	Ma03_p29790.1	SP-LRR-TM	1073	119.30	5.36	Cell membrane
*MaLRR-RLP23*	Ma03_p29840.1	SP-LRR-TM	1065	117.41	4.94	Cell membrane
*MaLRR-RLP24*	Ma03_p29850.1	LRR-TM	704	78.89	5.81	Cell membrane
*MaLRR-RLP25*	Ma03_p29910.1	LRR-TM	717	80.29	5.91	Cell membrane
*MaLRR-RLP26*	Ma03_p29970.1	SP-LRR-TM	1728	191.53	5.29	Cell membrane
*MaLRR-RLP27*	Ma03_p30010.1	LRR-TM	716	79.78	5.50	Cell membrane
*MaLRR-RLP28*	Ma03_p31870.1	SP-LRR-TM	287	30.02	5.40	Cell membrane
*MaLRR-RLP29*	Ma04_p06430.1	SP-LRR-TM	1092	118.88	5.18	Cell membrane
*MaLRR-RLP30*	Ma04_p07610.1	SP-LRR-TM	721	77.83	8.41	Cell membrane
*MaLRR-RLP31*	Ma04_p20720.1	SP-LRR-TM	841	92.49	5.50	Cell membrane
*MaLRR-RLP32*	Ma04_p22770.1	SP-LRR-TM	728	81.22	5.01	Cell membrane
*MaLRR-RLP33*	Ma04_p22780.1	SP-LRR-TM	943	104.62	5.64	Cell membrane
*MaLRR-RLP34*	Ma04_p29910.1	SP-LRR-TM	984	107.17	5.60	Cell membrane
*MaLRR-RLP35*	Ma04_p32290.1	SP-LRR-TM	599	66.07	7.07	Cell membrane
*MaLRR-RLP36*	Ma04_p32300.1	SP-LRR-TM	597	65.97	5.97	Cell membrane
*MaLRR-RLP37*	Ma04_p32760.1	SP-LRR-TM	974	108.41	5.11	Cell membrane
*MaLRR-RLP38*	Ma04_p32770.1	LRR-TM	1009	111.79	4.82	Cell membrane
*MaLRR-RLP39*	Ma04_p32780.1	LRR-TM	871	96.61	4.64	Cell membrane
*MaLRR-RLP40*	Ma04_p32810.1	SP-LRR-TM	993	110.20	4.74	Cell membrane
*MaLRR-RLP41*	Ma04_p33500.1	SP-LRR-TM	338	36.02	9.24	Cell membrane
*MaLRR-RLP42*	Ma04_p35850.1	SP-LRR-TM	961	103.93	5.12	Cell membrane
*MaLRR-RLP43*	Ma04_p36340.1	SP-LRR-TM	980	110.29	6.02	Cell membrane
*MaLRR-RLP44*	Ma05_p12870.1	SP-LRR-TM	854	94.06	8.66	Cell membrane
*MaLRR-RLP45*	Ma05_p16530.1	SP-LRR-TM	462	50.00	8.33	Cell membrane
*MaLRR-RLP46*	Ma05_p19050.1	LRR-TM	481	52.82	4.94	Cell membrane
*MaLRR-RLP47*	Ma06_p10770.1	SP-LRR-TM	982	107.99	6.20	Cell membrane
*MaLRR-RLP48*	Ma06_p10790.1	SP-LRR-TM	925	103.14	5.72	Cell membrane
*MaLRR-RLP49*	Ma06_p10800.1	SP-LRR-TM	1667	183.83	5.41	Cell membrane
*MaLRR-RLP50*	Ma06_p10810.1	LRR-TM	974	107.05	5.93	Cell membrane
*MaLRR-RLP51*	Ma06_p10820.1	SP-LRR-TM	990	109.57	5.94	Cell membrane
*MaLRR-RLP52*	Ma06_p10830.1	SP-LRR-TM	972	107.84	5.78	Cell membrane
*MaLRR-RLP53*	Ma06_p10880.1	SP-LRR-TM	475	51.97	5.45	Cell membrane
*MaLRR-RLP54*	Ma06_p26220.1	SP-LRR-TM	275	29.17	8.51	Cell membrane
*MaLRR-RLP55*	Ma06_p33510.1	SP-LRR-TM	427	45.03	8.64	Cell membrane
*MaLRR-RLP56*	Ma07_p04260.1	SP-LRR-TM	733	79.43	6.71	Cell membrane
*MaLRR-RLP57*	Ma07_p18320.1	LRR-TM	891	97.53	5.57	Cell membrane
*MaLRR-RLP58*	Ma07_p19540.1	SP-LRR-TM	1115	122.28	5.73	Cell membrane
*MaLRR-RLP59*	Ma07_p26290.1	SP-LRR-TM	465	49.96	5.06	Cell membrane
*MaLRR-RLP60*	Ma07_p28580.1	SP-LRR-TM	285	29.63	5.16	Cell membrane
*MaLRR-RLP61*	Ma08_p16720.1	SP-LRR-TM	484	52.98	5.32	Cell membrane
*MaLRR-RLP62*	Ma09_p00750.1	SP-LRR-TM	478	52.30	6.09	Cell membrane
*MaLRR-RLP63*	Ma09_p05420.1	SP-LRR-TM	424	44.62	8.48	Cell membrane
*MaLRR-RLP64*	Ma09_p22650.1	SP-LRR-TM	1013	112.09	5.28	Cell membrane
*MaLRR-RLP65*	Ma09_p22680.1	LRR-TM	1040	114.95	5.03	Cell membrane
*MaLRR-RLP66*	Ma09_p22690.1	LRR-TM	1082	119.59	5.07	Cell membrane
*MaLRR-RLP67*	Ma09_p22710.1	SP-LRR-TM	1336	148.51	5.11	Cell membrane
*MaLRR-RLP68*	Ma09_p22720.1	LRR-TM	587	65.49	5.64	Cell membrane
*MaLRR-RLP69*	Ma10_p00450.1	SP-LRR-TM	624	70.87	5.51	Cell membrane
*MaLRR-RLP70*	Ma10_p00470.1	LRR-TM	585	65.92	6.66	Cell membrane
*MaLRR-RLP71*	Ma10_p00480.1	LRR-TM	631	71.10	5.38	Cell membrane
*MaLRR-RLP72*	Ma10_p00490.1	SP-LRR-TM	677	76.94	6.39	Cell membrane
*MaLRR-RLP73*	Ma10_p00500.1	SP-LRR-TM	676	75.97	5.14	Cell membrane
*MaLRR-RLP74*	Ma10_p00550.1	SP-LRR-TM	902	100.82	5.43	Cell membrane
*MaLRR-RLP75*	Ma10_p00570.1	SP-LRR-TM	917	102.26	5.22	Cell membrane
*MaLRR-RLP76*	Ma10_p16560.1	SP-LRR-TM	722	77.99	4.99	Cell membrane
*MaLRR-RLP77*	Ma10_p20250.1	LRR-TM	449	49.50	5.28	Cell membrane
*MaLRR-RLP78*	Ma10_p28400.1	SP-LRR-TM	1034	112.83	5.72	Cell membrane

^a^ SP, signal peptide; LRR, leucine-rich repeat domain; TM, transmembrane domain.

**Table 2 genes-13-00638-t002:** Seven *MaLRR-RLP* genes proposed as Foc race 1 RGCs in the present study (highlighted in gray). These seven Foc race 1 RGCs were located near the distal part of chromosome 10 of *M. acuminata* ssp. *malaccensis*, where resistance to Foc race 1 has been mapped [48].

#	*MaLRR-RLP* Gene	Sequence ID	Domains ^a^	Protein Length	kDa	pI	Subcellular Location
1		Ma10_p00400.1	Kinase	179	19.80	5.29	Nucleus
2		Ma10_p00410.1	Malectin	397	44.11	8.62	Extracellular
3		Ma10_p00420.1	SP-LRR	377	42.13	6.13	Extracellular
4		Ma10_p00430.1	Kinase	128	14.25	6.28	Cytoplasm
5		Ma10_p00440.1	LRR	209	23.29	5.94	Nucleus
6	*MaLRR-RLP69*	Ma10_p00450.1	SP-LRR-TM	624	70.87	5.51	Cell membrane
7		Ma10_p00460.1	Kinase	138	15.77	5.91	Mitochondrion
8	*MaLRR-RLP70*	Ma10_p00470.1	LRR-TM	585	65.92	6.66	Cell membrane
9	*MaLRR-RLP71*	Ma10_p00480.1	LRR-TM	631	71.10	5.38	Cell membrane
10	*MaLRR-RLP72*	Ma10_p00490.1	SP-LRR-TM	677	76.94	6.39	Cell membrane
11	*MaLRR-RLP73*	Ma10_p00500.1	SP-LRR-TM	676	75.97	5.14	Cell membrane
12		Ma10_p00510.1	LRR	480	53.69	5.79	Cytoplasm
13		Ma10_p00520.1	Malectin	241	26.28	5.26	Cytoplasm
14		Ma10_p00530.1	SP-LRR	166	18.92	8.58	Extracellular
15	*MaLRR-RLP74*	Ma10_p00550.1	SP-LRR-TM	902	100.82	5.43	Cell membrane
16		Ma10_p00560.1	SP-LRR	348	38.59	5.84	Extracellular
17	*MaLRR-RLP75*	Ma10_p00570.1	SP-LRR-TM	917	102.26	5.22	Cell membrane
18		Ma10_p00650.1	LRR	641	71.47	5.62	Cytoplasm
19		Ma10_p00660.1	LRR-TM	143	16.84	5.47	Endoplasmic reticulum membrane
		*I* (Solyc11g011180) ^b^	SP-LRR-TM	994	111.79	5.56	Cell membrane
		*I-7* (AKR80573) ^c^	SP-LRR-TM	966	108.15	5.72	Cell membrane

^a^ SP, signal peptide; LRR, leucine rich repeat domain; TM, transmembrane domain.; ^b,c^ Tomato *LRR-RLP* genes that confer resistance to *F. oxysporum* f. sp. *lycopersici.*

**Table 3 genes-13-00638-t003:** Total number of banana *LRR-RLPs* proposed in this study as resistance gene candidates (in bold) based on their close phylogenetic relationships with known *LRR-RLP* genes and genetic mapping data of Foc race 1 resistance [48]. The role of known *LRR-RLPs* in plant immunity that are phylogenetically closely related to banana *LRR-RLPs* is shown. Additional information on *LRR-RLP* resistance genes is provided in Appendix A.

Clade Name	*LRR-RLP* Gene	Role in Plant Immunity
	*MaLRR-RLP58*	Resistance gene candidate
I-7	*Cf-2*	Resistance to fungus *Cladosporium fulvum*
	*Cf-4*	Resistance to fungus *Cladosporium fulvum*
	*Cf-5*	Resistance to fungus *Cladosporium fulvum*
	*Cf-9*	Resistance to fungus *Cladosporium fulvum*
	*Hcr9-4E*	Resistance to fungus *Cladosporium fulvum*
	*ELR*	Resistance to oomycete *Phytophthora infestans*
	*I-7*	Resistance to fungus *Fusarium oxysporum*
	*Ve1*	Resistance to fungi *Verticillium dahliae* and *V.* *albo-atrum*
	*RLP23*	Resistance to fungus *Botrytis cinerea*
	*RLP30*	Resistance to fungus *Sclerotinia sclerotiorum*
	*RBPG1*	Resistance to fungus *Hyaloperonospora arabidopsidis*
	*RLM2*	Resistance to fungus *Leptosphaeria maculans*
	*MaLRR-RLP78*	Resistance gene candidate
I	*I*	Resistance to fungus *Fusarium oxysporum*
	*CuRe1*	Resistance to parasitic plant *Cuscuta reflexa*
	*ReMAX*	Perception of the *Xanthomonas* protein eMAX
Foc race 1 RGCs	*MaLRR-RLP69*	Resistance gene candidate
	*MaLRR-RLP70*	Resistance gene candidate
	*MaLRR-RLP71*	Resistance gene candidate
	*MaLRR-RLP72*	Resistance gene candidate
	*MaLRR-RLP73*	Resistance gene candidate
	*MaLRR-RLP74*	Resistance gene candidate
	*MaLRR-RLP75*	Resistance gene candidate
HcrVf2	*MaLRR-RLP33*	Resistance gene candidate
	*HcrVf2*	Resistance to fungus *Venturia inaequalis*
	*EIX2*	Perception of a fungal xylanase (EIX)
RFO2	*MaLRR-RLP35*	Resistance gene candidate
	*MaLRR-RLP36*	Resistance gene candidate
	*MaLRR-RLP46*	Resistance gene candidate
	*MaLRR-RLP56*	Resistance gene candidate
	*RFO2*	Resistance to fungus *Fusarium oxysporum*

## Data Availability

Not applicable.

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
