# Peer review of "Genome-Wide Analysis of the LRR-RLP Gene Family in a Wild Banana (Musa acuminata ssp. malaccensis) Uncovers Multiple Fusarium Wilt Resistance Gene Candidates"

_genes, 2022, doi:10.3390/genes13040638_

Round 1

Reviewer 1 Report

Fusarium oxysporum f. sp. cubense (Foc) is threatening world banana production, the knowledge of global R genes identification and expression patterns influenced by infection of different Foc races will help us to understand the host responses to the infection. The authors identified 78 LRR-RLP genes in Musa acuminata ssp. malaccensis genome and found 7 valuable RGCs for Fusarium wilt. The study is interesting, but there are still a number of issues need to be addressed.

  1. In the section of 3.6. The authors performed RT-qPCR in leaf and root tissues of the wild banana acuminata ssp. burmannica. I think it is better to carry out this experiment in M. acuminata ssp. malaccensis, because the article is based on the identification of R genes in M. acuminata ssp. malaccensis genome.
  2. Line 187, total amount of RNA should be 5 µg.
  3. Line 183-184, in vitro should be italic ?
  4. Figure 7a, should be indicate where the data come from, even if it is stated in the text.
  5. Please provide the primers sequence used in RT-qPCR in the supplementary.

Reviewer 2 Report

Banana is one of the most popular fruit in the world, while the production is constantly threatened by several biotic stresses, including fusarium wilt. This study identified the LRR-RLP family in Banana and performed a systematic analysis of this family. Notably, several resistance gene candidates of fusarium wilt were raised in this study.

1. The banana LRR-RLP genes should be named according to their location information as the previous LRR-RLK identification study (Identification and characterization of LRR-RLK family genes in potato reveal their involvement in peptide signalling of cell fate decisions and biotic/abiotic stress responses. Cells, 7(9), 120).

2. All genes should be italic like the LRR-RLP gene, the 25s gene in Line196, as well as the Arabidopsis in Line 30.

3. Specify the e-value considered during the HMM search in Line 134.

4. The domain number of Pfam and Smart in Line141 should be listed.

5. In Line 171, authors should opt for the latest version of MEGA like MEGA 11.

6. In Line 200, it should be the 2-ΔΔct was adopted.

7. Several resistance gene candidates of fusarium wilt were raised by the genetic mapping data, while the expression profile of these genes seems to be absent. Please add the expression profile of these members by qRT-PCR under fusarium wilt, which provides more clues to support your conclusions.
